# Cutting-Edge: Preclinical and Clinical Development of the First Approved Lag-3 Inhibitor

**DOI:** 10.3390/cells11152351

**Published:** 2022-07-30

**Authors:** Luisa Chocarro, Ana Bocanegra, Ester Blanco, Leticia Fernández-Rubio, Hugo Arasanz, Miriam Echaide, Maider Garnica, Pablo Ramos, Sergio Piñeiro-Hermida, Ruth Vera, David Escors, Grazyna Kochan

**Affiliations:** 1Oncoimmunology Research Unit, Navarrabiomed-Fundación Miguel Servet, Universidad Pública de Navarra (UPNA), Hospital Universitario de Navarra (HUN), Instituto de Investigación Sanitaria de Navarra (IdiSNA), 31001 Pamplona, Spain; eblancop@navarra.es (E.B.); lfernanr@navarra.es (L.F.-R.); harasane@navarra.es (H.A.); mechaidg@navarra.es (M.E.); mgarnics@navarra.es (M.G.); pramosca@navarra.es (P.R.); sergio.pineiro.hermida@navarra.es (S.P.-H.); descorsm@navarra.es (D.E.); grkochan@navarra.es (G.K.); 2Division of Gene Therapy and Regulation of Gene Expression, Cima Universidad de Navarra, Instituto de Investigación Sanitaria de Navarra (IdISNA), 31001 Pamplona, Spain; 3Medical Oncology Unit, Hospital Universitario de Navarra (HUN), Instituto de Investigación Sanitaria de Navarra (IdiSNA), 31001 Pamplona, Spain; ruth.vera.garcia@navarra.es

**Keywords:** opdualag, relatimab, BMS-986016, nivolumab, LAG-3, PD-1

## Abstract

Immune checkpoint inhibitors (ICIs) have revolutionized medical practice in oncology since the FDA approval of the first ICI 11 years ago. In light of this, Lymphocyte-Activation Gene 3 (LAG-3) is one of the most important next-generation immune checkpoint molecules, playing a similar role as Programmed cell Death protein 1 (PD-1) and Cytotoxic T-Lymphocyte Antigen 4 (CTLA-4). 19 LAG-3 targeting molecules are being evaluated at 108 clinical trials which are demonstrating positive results, including promising bispecific molecules targeting LAG-3 simultaneously with other ICIs. Recently, a new dual anti-PD-1 (Nivolumab) and anti-LAG-3 (Relatimab) treatment developed by Bristol Myers Squibb (Opdualag), was approved by the Food and Drug Administration (FDA) as the first LAG-3 blocking antibody combination for unresectable or metastatic melanoma. This novel immunotherapy combination more than doubled median progression-free survival (PFS) when compared to nivolumab monotherapy (10.1 months versus 4.6 months). Here, we analyze the large clinical trial responsible for this historical approval (RELATIVITY-047), and discuss the preclinical and clinical developments that led to its jump into clinical practice. We will also summarize results achieved by other LAG-3 targeting molecules with promising anti-tumor activities currently under clinical development in phases I, I/II, II, and III. Opdualag will boost the entry of more LAG-3 targeting molecules into clinical practice, supporting the accumulating evidence highlighting the pivotal role of LAG-3 in cancer.

## 1. Introduction: Brief History of Immunotherapy

The use of monoclonal antibodies blocking T-cell inhibitory receptors has enormously revolutionized the treatment of various haematological and solid cancers, thanks to the reinvigoration of endogenous antitumor immune responses. Since its early beginnings, immunotherapy has rapidly evolved, leading to durable clinical responses in patients with advanced-stage tumors. Nowadays, immune checkpoint blockade (ICB) constitutes a standard clinical therapy, even as a first-line choice. Up to date, nine immune checkpoint inhibitors (ICIs) targeting Programmed cell Death protein 1 (PD-1), Programmed cell death ligand 1 (PD-L1), Cytotoxic T-Lymphocyte Antigen 4 (CTLA-4) and Lymphocyte-Activation Gene 3 (LAG-3) receptors have been approved for clinical use. Despite its enormous potential, ICB immunotherapy still presents a variety of drawbacks that remain to be addressed. The failure to respond in a significant number of patients is probably the main one. This failure can be attributed to tumor refractoriness, acquired resistance, and deleterious immune-related adverse events. Aiming to increase the efficacy of ICB monotherapies, several combinatorial approaches have been developed. The rationale behind these combinations is the targeting of differential mechanisms exerted by distinct checkpoint receptors and their corresponding ligands in the tumor microenvironment. This co-targeting is thought to trigger synergistic immune responses in terms of increased progression-free survival (PFS) and reduced number of unresponsive patients. On 18 March 2022, the Food and Drug Administration (FDA) approved one of these combinatorial immunotherapies. Under the name of Opdualag, the combination of the LAG-3 and PD-1 ICIs relatlimab and nivolumab gained authorization for use in adult and some paediatric patients with unresectable or metastatic melanoma. Here, we will contextualize the clinical trial that brought this therapy to the market, RELATIVITY-047. We will review the pathway of Opdualag towards its approval. In addition, we will summarize the results achieved by the other LAG-3 targeting molecules currently at phase I, I/II, II and III, and review the preclinical and clinical development of other ‘second generation’ ICIs that may follow the steps of LAG-3 in a near future.

## 2. Preclinical and Clinical Development of Opdualag

### 2.1. LAG-3 Molecular Function

The LAG-3 molecule has recently emerged as a promising cancer immunotherapy target and a highly important next-generation immune checkpoint molecule [1,2,3]. Closely related to CD4, and adjacent to its locus, it presents a similar genetic organization [4]. LAG-3 was first described as an immune inhibitory receptor in activated T cells. It plays a similar role than its immune-checkpoint counterparts PD-1 and CTLA-4 [5,6]. LAG-3 exerts an inhibitory function over multiple biological functions, such as T cell activation, immune function, proliferation, cytokine secretion, effector functions and T cell homeostasis [5,6,7,8,9]. For example, LAG-3 regulates the size of the expanding T cell population following antigen activation in vivo [8]. In broad terms, LAG-3 down-modulates TCR:CD3 intracellular signal transduction cascades and calcium fluxes within the immunological synapse, terminating cytokine and T cell responses to the TCR:CD3 activation, while favouring CD4 and CD8 T cell exhaustion [8,10,11,12,13,14,15].

LAG-3 is expressed by many T cell subsets, including: CD4 T helper cells, cytotoxic CD8 T cells, activated T cells, NK T cells, effector CD4 T cells, regulatory T cells, CD8 tumor-infiltrating lymphocytes and tumor-infiltrating antigen-specific CD8 T cells [4,16,17,18,19,20,21,22,23,24,25,26,27]. However, LAG-3 expression is not limited to T cells, having been described in B cells, natural regulatory plasma cells or B cells, plasmacytoid dendritic cells (DCs) and in neurons among others [15,27,28,29,30,31]. 

Initially, its high-affinity binding with the class II Major Histocompatibility Complex (MHC-II) was considered to mediate its inhibitory functions. MHC-II was thought to be its canonical ligand. MHC-II binds to LAG-3 with higher affinity than CD4, thus inhibiting CD4 T cell activation by competition with its binding to CD4 [5,6,32,33,34,35,36]. However, while it is undeniable that LAG-3:MHC-II interaction plays a critical role, LAG-3 binding to other ligands contributes to its inhibitory activities. The next functional ligands to be described were galectin-3 (Gal-3), critical to inhibit T cell activation and CD8 cytotoxic T cell functions [37,38,39,40], the liver-secreted protein fibrinogen-like protein 1 (FGL1), critical for tumor immune evasion mechanisms in response to anti-PD-1/anti-PD-L1 treatments [9,37,41,42,43], and the DC-specific Intercellular adhesion molecule-3-grabbing non-integrin family member (LSECtin) in melanoma cells, inhibiting cyclin-dependent kinases [44].

### 2.2. LAG-3 Clinical Research

LAG-3 is generally considered an aggressive progression marker in several haematological and solid malignancies, driving T cell exhaustion and pro-apoptosis, and associated with poor prognosis and decreased survival. Its expression is also considered an intrinsic resistance mechanism to anti-PD-1/anti-PD-L1 therapies through its synergic co-expression with PD-1 [12,22,45,46,47,48,49,50,51,52]. For example, non-small cell lung cancer (NSCLC) patients who did not respond to anti-PD-L1/PD-1 monotherapies had highly dysfunctional T cells that strongly co-expressed PD-1 and LAG-3 after TCR stimulation [22]. ICB therapies, especially anti-PD-1/anti-PD-L1 treatments, have revolutionized cancer treatment in recent years. However, not all patients respond to treatment due to intrinsic or extrinsic resistance mechanisms [1]. LAG-3 over-expression confers resistance to PD-1 blockade. Indeed, PD-1/LAG-3 co-blockade is demonstrating encouraging results and strong capacities both in preclinical and clinical research [2]. In this context, LAG-3-targeted therapies have emerged as a cancer immunotherapy alone and in combination with anti-PD-1 treatments. A new generation of novel bispecific molecules is being evaluated with encouraging results in preclinical and clinical studies.

108 interventional clinical trials are evaluating 19 different LAG-3 targeting molecules in 39 phase I trials, 2 phase I/II trials, 40 phase II trials, 3 phase II/III trials and 3 phase III trials (Appendix A). These molecules can be divided into anti-LAG-3 monoclonal antibodies (178 trials, 10 molecules), bispecific molecules (14 trials, 7 molecules), LAG-3 fusion proteins (15 trials, 2 molecules) and CAR-T cells (1 trial, 1 molecule). In total, more than 28,000 adult patients are being enrolled with the exception of the NCT03470922 phase II/III trial, which is enrolling patients of 12 years old patients and older. A total of 23 trials are active but not recruiting, 16 are completed, 8 not yet recruiting, 49 recruiting, 9 terminated and 2 withdrawn. Only 13 have available results. According to allocation, 50 trials are randomized, 39 non-randomized, and 19 N/A. On the intervention model, 68 trials follow a parallel assignment, 1 a crossover assignment, 25 a single group assignment and 13 a sequential assignment. Most of them follow an open label masking, while three of them a single (participant), four a double (participant and investigator), two a triple (participant, care provider and investigator) and four a quadruple (participant, care provider, investigator and outcomes assessor) (Figure 1 and Figure 2). Treated neoplasias include most hematological and solid cancers, but also psoriasis, ulcerative colitis and influenza.

Interestingly, LAG-3 expression is associated with increased pathology and impaired immune responses in multiples diseases such as Parkinson’s Disease [29,30], cardiovascular diseases (increased coronary heart disease and increased myocardial infarction) [53,54], HDL Hypercholesterolemia [55,56], Inflammatory Bowel Disease [57,58], Multiple Sclerosis [59], Diabetes Mellitus [60,61] and infection (Salmonella [31], Plasmodium parasites (P. yoelii 17XL, P yoelii 17XNL, P. chabaudi, P. vinckei, and P. berghei) [62], Mycobacterium tuberculosis [63], human immunodeficiency virus (HIV) [64], non-pathogenic simian immunodeficiency virus (SIV) [65], hepatitis B virus (HBV) [66], human papillomavirus (HPV) [67], hepatitis C virus (HCV) [67], lymphocytic choriomeningitis viral (LCMV), herpes simplex virus 1 (HSV-1) and other chronic viral infections [15,68,69,70,71,72]. Thus, LAG-3 targeted strategies currently under clinical development for cancer will also be relevant as immunotherapies for the treatment of non-neoplastic diseases [73,74,75,76,77].

#### 2.2.1. Anti-LAG-3 Monoclonal Antibodies

78 clinical trials are evaluating 10 different anti-LAG-3 monoclonal antibodies: BMS-986016 or relatlimab (Bristol-Myers Squibb), GSK2831781 (GlaxoSmithKline), HLX26 (Fosun Pharma), IBI110 (Innovent Biologics), INCAGN02385 (Incyte), LAG525 or IMP701 (Novartis), MK-4830 or favezelimab (Merck), REGN3767 or fianlimab (Regeneron Pharmaceuticals and Sanofi), Sym022 (Symphogen), TSR-033 (Tesaro) (Appendix A). Of these, 21 are phase I, 19 phase I/II, 24 phase II, 1 phase II/III and 3 phase III. 7 phase I and II trials are only evaluating the anti-LAG-3 treatment alone (NCT05078593, NCT03489369, NCT03965533, NCT02195349, NCT05039658, NCT03538028, NCT03893565), 10 phase I, I/II and III trials are evaluating the anti-LAG-3 treatment alone and in combination with other ICIs (NCT02658981, NCT03250832, NCT03005782, NCT02966548, NCT02720068, NCT04150965, NCT01968109, NCT02061761, NCT02460224, NCT03743766) and the rest are evaluating anti-LAG-3 treatments in combination with other treatments, mainly with additional ICIs such as anti-PD-1. Most of them are fully humanized IgG4 blocking antibodies.

**BMS-986016** or **relatlimab**, developed by Bristol-Myers Squibb in 47 clinical trials, was the first anti-LAG-3 monoclonal antibody to be clinically developed and the first one to receive the FDA approval for its clinical use. It has 4 subunits, with 16 disulfide links and 2 N-glycosylation sites, with an average molecular weight of 145.3 kDa [78]. Phase I (7 trials), I/II (12 trials), II (26 trials), II/III (1 trial) and III (1 trial) preliminary results showed good tolerability, efficacy, toxicity and antitumour profiles alone or in combination with anti-PD-1/PD-L1 blockade immunotherapies, as a good alternative to overcome immunotherapy resistance [79,80,81]. For example, it restores T cell mediated responses and TNF-a, IFN-y and IL-2 cytokine release [82]. The phase III clinical trial that led to the LAG-3/PD-1 combination approval for melanoma treatment is further discussed in the next section.**GSK2831781**, derived from **IMP731** Immunetep’s antibody, developed in monotherapy by GlaxoSmithKline in 3 clinical trials (2 phase I and 1 phase II) for psoriasis and ulcerative colitis. The ulcerative colitis phase II trial was terminated after an interim analysis [83], but phase I results show good tolerability, safety and inflammation regulation profiles [84].**HLX26**, developed by Fosun Pharma in 2 phase I clinical trials (NCT05078593 and NCT05400265), where its safety, tolerability, pharmacokinetic characteristics and preliminary efficacy are being evaluated alone and in combination with anti-PD-1 treatments in patients with solid tumors or lymphoma.**IBI110**, developed by Innovent Biologics in a phase I clinical trial alone and in combination with anti-PD-1 in patients with relapsed or refractory diffuse large B cell lymphoma (r/r DLBCL) (NCT05039658).**INCAGN02385**, is being developed by Incyte in 4 clinical trials (1 phase I, 1 phase I/II and 2 phase II) alone (NCT03538028) or in combination (NCT04370704, NCT05287113, NCT04586244) with anti-PD-1 and anti-TIM-3 immune checkpoint therapies. Preliminary data shows good tolerability profiles [85].**LAG525** or **IMP701** developed by Novartis in 5 clinical trials (1 phase I, 1 phase I/II and 3 phase II), alone or in combination with anti-PD-1 blockers. The structure of this antibody consists of 4 subunits, 16 disulfide bridges and 2 N-glycosylation sites, with an estimated molecular weight of 147 kDa [86]. Preliminary data demonstrate promising pharmacokinetics, antitumour activity and safety alone and in combination [87,88,89].**MK-4830** or **favezelimab**, developed by Merck in 8 clinical trials (1 phase I, 5 phase I/II, 1 phase II and 1 phase III), alone or in combination with anti-PD-1, oxaliplatin, Leucovorin (Calcium Folinate), Fluorouracil [5-FU] and lenvatinib, showing manageable safety and tolerability alone and in combination. In fact, anti-LAG-3/anti-PD-1 combination showed a 6.3% objective response rate, better than the monotherapy treatment, with similar treatment-related adverse effects [90,91]. The structure of this antibody consists of 4 subunits, 16 disulfide bridges and 2 glycosylation sites, with an estimated molecular weight of 146 kDa [92].**REGN3767** or **fianlimab**, developed by Regeneron Pharmaceuticals and Sanofi in 3 clinical trials (1 phase I, 1 phase II and 1 phase III), promotes T cell activation and T cell mediated cytotoxicity with good pharmacokinetics and toxicology profiles in vitro and in vivo [93]. The structure of the antibody is composed of 4 subunits, 16 sulfide bridges and 2 N-glycosilation sites [94]. Early efficacy and antitumor activity were also suggested in the preliminary clinical trials results. Its combination with cemiplimab also showed a good safety profile [95,96,97]. The combination with anti-PD-1, and cemiplimab is being evaluated in phase I (NCT03005782), II (NCT01042379) and III (NCT05352672) trials while it is being studied alone in the NCT03005782 phase I trial. Interestingly, anti-LAG-3 PET tracers (89Zr-DFO-REGN3767) are being clinically developed to establish the tracer biodistribution and dosimetry, monitoring the response to REGN3767 treatment (NCT05346276, NCT04706715, NCT04566978). However, these clinical trials are not being considered in this review as LAG-3 targeting clinical trials, because their main purpose is establishing PET scanning as a diagnostic method.**Sym022**, developed by Symphogen in 3 phase I clinical trials, is being evaluated for dose-escalation and dose-expansion alone (NCT03489369) or in combination (NCT03311412, NCT04641871) with anti-PD-1 and anti-TIM-3 immunotherapies. The treatment combination showed synergic antitumor activity in preclinical models [98,99].**TSR-033**, is being developed by Tesaro in 2 phase I clinical trials, alone and in combination with anti-PD-1 and anti-TIM-3 treatments (NCT03250832, NCT02817633). The combination with PD-1 blockers increases CD4 T cell activation and IL-2 production and cell proliferation [100]. Phase I preliminary data indicates good safety and tolerability.

#### 2.2.2. Anti-LAG-3 Bispecific Antibodies 

**ABL501** is being developed by ABL Bio in a phase I trial for the treatment of any progressive, locally advanced (unresectable) or metastatic solid tumor (NCT05101109). This bispecific antibody blocks PD-L1 and LAG-3 as a single agent. Dose-escalation analysis is being conducted. The dosing interval to be used in the dose-expansion part will be re-evaluated based on the emerging safety and pharmacokinetics data from the dose-escalation part of the study. It promotes enhanced human T cell activation in vitro and potentiates antitumor responses of T cells through DC activation [101,102].**IBI323,** a LAG-3/PD-L1 bispecific antibody, is being developed by Innovent Biologics in a phase I clinical trial alone and in combination with chemotherapy in patients with advanced malignancies. The purpose of this study is to evaluate IBI323 safety, tolerability and efficacy. It enhances tumor-specific immunity in vitro [103].**MGD013** or **Tebotelimab**, a LAG-3/PD-1 bispecific DART^®^ antibody, is being developed by MacroGenetics in 7 clinical trials (3 phase I, 1 phase I/II, 1 phase II and 2 phase II/III) in patients with unresectable or metastatic neoplasms (NCT03219268), patients with advanced or metastatic solid tumors who failed prior treatment (NCT04178460), melanoma (NCT04653038), liver cancer (NCT04212221), Head and Neck Cancer (NCT04634825, NCT04082364) and HER2+ Gastric/GEJ Cancer (NCT04082364), to evaluate its safety and efficacy, alone or in combination with margetuximab (anti-HER2), niraparib (a selective PARP1/2 inhibitor), Brivanib Alaninate (Multitargeted tyrosine kinase inhibitor) and enoblituzumab (Anti-B7-H3 antibody). Preliminary results showed good tolerability, safety and antitumour activity profiles [104].**RO7247669**, a LAG-3/PD-1 bispecific antibody, is being developed by Hoffmann-La Roche in 1 phase I and 1 phase II clinical trials in patients with advanced and/or metastatic solid tumors (NCT04140500) and advanced or metastatic squamous cell carcinoma of the oesophagus (NCT04785820), alone or in combination with a PD-1/TIM-3 bispecific antibody or an anti-PD-1 single agent.**XmAb^®^22841** or **pavunalimab**, a LAG-3/CTLA-4 bispecific antibody, is being developed by Xencor in a phase I clinical trial (NCT03849469), alone and in combination with anti-PD-1 as a single agent in selected advanced solid tumors. It enhances antitumor activity, T cell activation, cytokine secretion and cell proliferation [105].**EMB-02**, a LAG-3/PD-1 bispecific antibody, is being developed as a single treatment agent by EpimAb Biotherapeutics in a phase I/II clinical trial (NCT04618393) in advanced solid tumors. Dose escalation followed by cohort expansion will be performed. In vivo preclinical data showed antitumor activity in anti-PD-1 resistant models.**FS118**, a LAG-3/PD-L1 bispecific antibody, is being developed as a single agent treatment by F-star Therapeutics in a phase I/II clinical trial (NCT03440437) in patients with advanced malignancies, to determine dosing and toxicity. It enhanced T cell activation and antitumor activity in vitro and in vivo [105,106,107]. Preliminary clinical trial data showed good pharmacodynamics and tolerability profiles. [108,109].**CB213** Humabody^®^, a PD-1xLAG-3 antagonist developed by Crescendo Biologics Ltd., have recently entered a partnership with Cancer Research UK for its clinical development into a future phase I clinical trial ([110]). This bispecific molecule binds and blocks with high affinity PD1 and LAG-3 on PD-1+LAG-3+ T cells, induces ex vivo T cell proliferation of dysfunctional T cells from NSCLC patients, with superior activity than anti-PD-1 alone and suppress tumor growth in vivo [111].

#### 2.2.3. LAG-3 Fusion Proteins

Two different LAG-3 fusion proteins are being developed in several phase I (9), I/II (2), and II (4) trials: IMP321 or Eftilagimod Alpha or Efti (Immutep) and EOC202 (Taizhou EOC Pharma) (Appendix A). EOC202 is a recombinant human LAG-3 fusion protein injection combined with albumin-bound paclitaxel for the treatment of patients with HR+, HER2- advanced breast cancer with progression after endocrine therapy (NCT05322720). On the other hand, IMP321 is the only soluble recombinant form of LAG-3 that is being clinically developed. In fact, IMP321 was the first LAG-3 targeted molecule to be studied in the clinic in 2006.

**IMP321**, Eftilagimod Alpha or Efti, a LAG-3 soluble fusion protein, is being developed by Immutep in 14 clinical trials (9 phase I, 2 phase I/II and 3 phase II) for the treatment of advanced solid tumors, hepatitis B and flu. IMP321 is being developed as an adjuvant and immune modulator for cancer and vaccines against infectious diseases, as well as an anticancer treatment agent. It is being tested alone and in combination with chemotherapy (gemcitabine), anti-PD-L1, anti-PD-1, paclitaxel, hepatitis B antigen (without alum), a reference flu antigen, Melan-A VLP vaccine and melanoma tumor-specific peptides. Data shows that IMP321 enhances T cell activation and proliferation, humoral, effector and adaptive immunity, cytokine release, immunogenicity and antitumor activity, with good tolerability, efficacy and safety profiles [112,113,114,115,116,117,118,119,120].**EOC202**, a recombinant human LAG-3 fusion protein, is being developed by Taizhou EOC Pharma in a phase II clinical trial (NCT05322720) in HR+, HER2- advanced breast cancer with progression after endocrine therapy to evaluate the PFS for EOC202 combined with albumin-bound paclitaxel versus albumin-bound paclitaxel alone.

#### 2.2.4. Anti-LAG-3 CAR-T Cells

One phase I/II clinical trial (NCT05410717) is evaluating Claudin6 targeting CAR-NK cells for Stage IV Ovarian Cancer, refractory testis cancer and recurrent endometrial cancer (Appendix A). To enhance the killing capability, some CAR-NK cells in this trial are genetically engineered to express and secret IL7/CCL19 and/or SCFVs against PD1/CTLA4/Lag3. The one study arm evaluates engineering Claudin6 targeting CAR combined with/or without IL7/CCL19 and/or scfv against PD1/CTLA4/Lag3 secreting vector into NK cells, which are isolated from patients with advanced ovarian cancer or other cancers with expression of Claudin6, and then transfusing them back the patients.

### 2.3. Opdualag and Its Pathway towards the Clinic

On 18 March2022, the FDA approved Opdualag as a first line treatment for unresectable or metastatic melanoma at a fixed dose combination. This approval signified a major historical achievement for Bristol-Myers Squibb, and a remarkable milestone for the landscape of cancer treatments. This therapeutic strategy established for the first time a next-generation LAG-3 blocker for clinical use. Opdualag consists of a pre-mixed combination of two IgG4 kappa monoclonal antibodies, nivolumab 480 mg (anti-PD-1,146 kDa) and relatlimab (BMS-98621) 160 mg (anti-LAG-3, 148 kDa) both expressed in Chinese Hamster Ovary (CHO) cell lines. This combination is prepared and provided to the patient through intravenous (IV) infusions every 4 weeks until disease progression or unacceptable toxicity occurs [121]. Its list price is $13,694.27, and it is indicated for adults and paediatric patients over the age of 12 with unresectable or metastatic melanoma that has spread or cannot be removed by surgery. 

Nivolumab (opdivo) is an anti-PD-1 monoclonal antibody from Bristol-Myers Squibb clinically developed in more than 35,000 patients including Phase 3, for the treatment of a variety of tumor types. Currently approved in more than 65 countries (including the United States, the European Union, Japan and China) Opdivo was the first anti-PD-1 immune checkpoint blocker to be approved for clinical use in July 2014. Later in 2015, the combination of nivolumab with ipilimumab (Yerboy), a CTLA-4 blocking monoclonal antibody, was approved for metastatic melanoma, demonstrating to be safe and effective. In addition, preclinical and clinical studies showed that nivolumab and relatlimab combination reactivated T and NK cell-mediated responses, enhanced T cell activation and cytokine production, restoring the effector functions of exhausted T cells [82].

The FDA Oncology Center of Excellent conducted a Project Orbis review on the novel drug in collaboration with the Australian Therapeutic Goods Administration (TGA), and Switzerland’s Swissmedic to provide a review of the oncology drugs framework among international partners. This was carried out using the Real-Time Oncology Review (RTOR) pilot program to streamline data submission prior to the clinical application filing and the Assessment Aid (an applicant voluntary submission). The application by Bristol-Myers Squibb (sponsor, study director, and responsible party of the trial) was then granted priority review, fast track, and orphan drug designation [122]. 

Opdualag was clinically evaluated in RELATIVITY-047 (NCT03470922), an interventional multi-institutional (127 locations all over the glove), randomized (1:1), parallel assignment interventionist, quadruple-blinded (participant, care provider, investigator, outcomes assessor) phase II/III trial that enrolled 714 patients (>12 years of age). The actual study start date was 11 April 2018, the primary completion date was 25 January 2021 and the estimated study completion date is 30 November 2023. The purpose of this study is to determine whether relatlimab in combination with nivolumab is more effective than nivolumab as a monotherapy in treating unresectable melanoma or metastatic melanoma. 

Inclusion criteria were histologically confirmed Stage III (unresectable) or Stage IV melanoma per the AJCC staging system, not having had prior systemic anticancer therapy for these cancers, and that tumor tissue from an unresectable or metastatic site of disease must be provided for biomarker analyses. The two sexes were eligible for the study. Participants must have a documented BRAF status prior to randomization. Exclusion criteria were that participants must not have active brain metastases or leptomeningeal metastases, uveal melanoma, nor active, known, or suspected autoimmune disease. Healthy volunteers were not accepted. No lifestyle restrictions were required during treatment.

The primary outcome in the trial was PFS determined by Blinded Independent Central Review (BICR) using RECIST v1.1 from randomization to date of first documented tumor progression or death (up to approximately 33 months). The secondary outcome were Overall Survival (OS) and Overall Response Rate (ORR), from randomization to the date of death (up to approximately 3 years). Other outcome measures were the number of participants experiencing adverse events (AEs), serious adverse events (SAEs), AEs leading to discontinuation, laboratory abnormalities in specific liver and thyroid tests from first dose to 30 days after last dose of the study (up to approximately 33 months), as well as the number deaths from first dose up to approximately 33 months. 

Patients received IV treatment (nivolumab 480 mg and relatlimab 160 mg) every 4 weeks (*n* = 355) or nivolumab 480 mg by IV infusion every 4 weeks (*n* = 359) (clinical protocol CA224047) [123]). The median treatment duration was 6 months (0–31 months range) in Opdualag-treated patients and 5 months (0–32 months range) in nivolumab-treated patients [124]. To compare the side effect profile of PD-1 monoblockade (Nivolumab arm) *versus* dual therapy (Opdualag arm), the most common (>10%) AEs and SAEs (>1%) are presented in Table 1. Briefly, opdualag potentially breaks peripheral tolerance and induces immune-mediated adverse reactions (IMARs). It can cause serious side effects. Indeed, 0.8% of the Opdualag-treated patients (3 patients) presented fatal AEs, 18% of the participants discontinued from Opdualag treatment due to AEs, and 43% of them had AEs that required dosage interruptions. In addition, the treatment can cause reproductive toxicity to pregnant women. No safety or effectiveness differences were observed upon the age of patients.

Opdualag demonstrated a statistically significant improvement in PFS when compared to nivolumab alone (HR = 0.75; 95% [CI]: 0.62, 0.92; *p*-value = 0.0055) with a PFS median of 10.1 months for the Opdualag arm (95% CI: 6.4, 15.7) versus 4.6 months in the nivolumab arm (95% CI: 6.4, 15.7). Opdualag did not showed a significant improvement in OS when compared to nivolumab alone (HR = 0.80; 95% CI: 0.64, 1.01) with a not reached OS median for Opdualag arm (95% CI: 34.2, NR) versus 34.1 months in the nivolumab arm (95% CI: 25.2, NR) [122,125,126,127].

## 3. Behind the Steps of LAG-3 towards the Clinic

The rationale behind the blockade of multiple immune checkpoint receptors relies on their varied mechanisms of action, whose combination may converge to a synergistic effect. Here, three main mechanistic pathways are addressed [128,129]:Immune activation at lymph nodes and peripheral tissues. The activation of T cells requires not only the antigen presentation through the MHC complex, but also a second costimulatory signal provided by APCs. T cells are initially primed at lymph nodes, although the interaction with other immune cell populations in peripheral tissues as well as in the tumor microenvironment may also provide immunoregulatory signals. These signals mediate the acquisition of effector functions or immunosuppressive phenotypes by T cells depending on the engagement of costimulatory or checkpoint receptors. In this case, immune checkpoint blockade would enhance T cell activation by APCs.Priming of immune tolerogenic phenotypes. Some immune checkpoint receptors induce tolerogenic phenotypes on antigen-presenting cells (APCs) and prime Tregs.Induction of T-cell dysfunction. Sustained antigen presentation and stimulation with inflammatory cytokines induce T cell exhaustion, characterized by reduced proliferation and effector functions. However, cytotoxic functions may be rescued by immune checkpoint receptors blockade with monoclonal antibodies. It has been reported that exhausted T cells show increased expression of multiple checkpoint receptors, which may interfere with the response rate of patients to ICB monotherapies.

Whereas the blockade of the PD-1/PD-L1 axis has focused attention on the ICB field due to its impressive and durable clinical responses, a plethora of immune checkpoint receptors have been identified in the last years, which are gaining attention. Apart from the classical checkpoints CTLA-4, PD-1, and PD-L1, the so-called second generation of ICIs targeting alternative receptors are gaining increasing relevance in their pathway toward their clinical application [130]. Some years ago, LAG-3 was considered one of these promising ‘next generation’ checkpoints. Importantly, after the approval of Opdualag by the FDA for its clinical use, this checkpoint has jumped to the first line of immunotherapeutic agents. Thus, LAG-3 has paved the way for other checkpoints. We review below the pre-clinical degree of development of the main ‘second generation’ checkpoint inhibitors that may follow the steps of LAG-3 in a near future.

### 3.1. TIM-3

TIM-3 (T-cell immunoglobulin and mucin domain-containing protein 3) is expressed by several cell populations, including CD4 and CD8 T cells [131], Tregs [132], myeloid cells [133], and NKs [134]. Its known ligands are galectin-9 [135], CEACAM-1 [136], phosphatidylserine [137], and HMGB1 [138], which interact with different regions of the TIM-3 extracellular immunoglobulin V domain, leading to the activation of differential signaling cascades. TIM-3 engagement would be mediated by differential expression of each ligand in each tissue microenvironment. In the context of cancer, TIM-3 expression has been found to be a marker of CD8 tumor-infiltrating T lymphocytes (TILs) dysfunctionality and Treg expansion, and correlates with tumor progression [139]. Consequently, the TIM-3 blockade restores antitumor CD8 T cell responses [140] and mediates Treg depletion [141]. Importantly, TIM-3 and PD-1 are usually co-expressed both in CD4 and CD8 T cells, being these cells highly dysfunctional in terms of impaired production of effector IFNγ, TNFα, and IL-2, thus promoting effector T cells exhaustion [140]. Consistently, a co-blockade of PD-1 and TIM-3 leads to a stronger reinforcement of T cell responses compared to PD-1 monoblockade, as shown in preclinical models [142,143,144]. TIM-3 as a single immunotherapy has shown limited results in clinical trials. Nevertheless, synergistic effects are boosted by the combination of anti-TIM-3 monoclonal antibodies with other ICIs, such as CTLA-4 and PD-1. Moreover, opposing these classical checkpoints, TIM-3 expression is restricted to terminally differentiated T cells and intratumoral T regs, thus avoiding broad AEs elicited by CTLA-4 [145] and PD-1 blockade [146,147].

Many ongoing clinical trials aim to analyze the co-blockade of TIM-3 and PD-1 (Appendix A). Most of the therapies designed to target TIM-3 are fully humanized IgG4 or IgG1 blocking antibodies that bind with high affinity to TIM-3. Moreover, several bi-specific antibodies for the dual blockade of PD-1 and TIM-3 have also been developed. However, only the anti-TIM-3 antibody cobolimab (TSR-022) has progressed to a phase II clinical trial up to date. TSR-022 is a humanized IgG4 monoclonal antibody against TIM-3, developed by Tesaro and GlaxoSmithKline. Early results from preclinical studies revealed that cobolimab enhanced IL-2 production by CD4 T cells in MLR assays in vitro, while in vivo studies demonstrated a good tolerability profile. Altogether, these results encouraged its evaluation in clinical assays [147,148]. Following preliminary efficacy results from the phase I AMBER study (NCT02817633) [149], the combination of cobolimab with the anti-PD-1 inhibitor dostarlimab (TSR-042) showed preliminary anti-tumor activity with acceptable tolerability in a range of advanced solid tumors including NSCLC, melanoma and peritoneal mesothelioma. The second phase of the study will evaluate the safety and efficacy of the combination of cobolimab with the anti-PD-1 blocking antibody dostarlimab in disease-specific cohorts including patients with NSCLC who received previous PD-L1 blockade immunotherapy and colorectal cancer patients with previous treatments.

### 3.2. TIGIT

TIGIT (T cell immunoreceptor with Ig and ITIM domains) is a member of the PVR-nectin family. It is expressed on CD4, CD8, γδ T cells, Treg and NKs. Its main ligand is PVR, which is usually overexpressed on tumor cells and tumor-associated myeloid cells [150,151], although other proteins, such as CD112 and CD113, also interact with this receptor. Its ligation suppresses T cell activation [152,153]. The mechanism of action of the TIGIT-PVR axis differs from the classical immune checkpoints, thus making it an attractive target for immunotherapy. Unlike PD-1 and CTLA-4, NKs constitutively express TIGIT, and its expression on intratumoral NKs is upregulated [154]. Therefore, anti-TIGIT treatments show the particular ability to target at the same time the two main antitumor effector populations, T lymphocytes, and NKs. Moreover, TIGIT blockade abrogates MDSCs-mediated NK suppression [155,156] and immunosuppressive activity of Tregs displaying a high TIGIT expression [157]. Since this checkpoint receptor establishes an immunosuppressive tumor microenvironment, its overexpression correlates with poor clinical outcomes [158,159,160,161]. Preclinical data in murine models demonstrate that TIGIT blockade resulted in reduced tumor growth and prolonged survival rates [154,162]. However, the co-blockade of TIGIT and PD-1/PD-L1 is required to achieve clinical responses in a therapeutic setting [152,154,163,164].

Several dozens of anti-TIGIT blocking antibodies or bispecific antibodies against PD-1/TIGIT are currently undergoing clinical trials in diverse stages of development (Appendix A). Most of them are IgG1 isotype antibodies that interact with Fcγ receptors. Among them, the most promising candidates to reach the market up to date are ociperlimab, tiragolumab, and vibostolimab, all of them under evaluation in phase III clinical trials.

Ociperlimab is a fully humanized IgG1 antibody developed by BeiGene that binds TIGIT with high affinity and specificity thanks to its intact IgG Fc function [165]. The combination of ociperlimab plus the anti-PD-1 antibody tislelizumab (BGB-A317) is currently under evaluation in patients diagnosed with distinct advanced or metastatic tumors, including NSCLC, esophageal squamous cell carcinoma, and cervical cancer. Preliminary results showed that this combination was well tolerated, showed antitumor activity, and elicited AEs consistent with tislelizumab monotherapy [166]. The phase III clinical trial AdvanTIG-302 (NCT04746924), sponsored by the pharmaceutical company BeiGene, is a multicentre, international, randomized, and double-blinded study analyzing the combo ociperlimab plus tislelizumab for the first-line treatment of patients with locally advanced, unresectable, or metastatic NSCLC whose tumors exhibit high PD-L1 expression and do not harbor EGFR-sensitizing mutations or ALK translocations.

Tiragolumab is a fully humanized IgG1 antibody developed by Hoffmann-La Roche and Genentech. It is the most advanced anti-TIGIT therapy, currently in late-stage clinical development. The combination of tiragolumab with the anti-PD-L1 antibody atezolizumab (Tecentriq^®^) was evaluated in the phase II clinical trial CITYSCAPE [167], whose preliminary results showed an 18% higher ORR compared to atezolizumab alone treated group, while the risk of death was reduced by 38%. Based on these positive data, tiragolumab received breakthrough therapy designation by FDA [168] for first-line treatment of NSCLC patients with high tumor PD-L1 expression and no EGFR or ALK mutations, thus making it an excellent candidate to reach the market in a near future. The ongoing phase III clinical trial SKYSCRAPER (NCT04294810) is currently evaluating the combo tiragolumab plus atezolizumab. Recent results indicated that PFS end-points of the combination arm with chemotherapy were not reached [169], while the study continues until mature OS data from the rest of the trial arms will be available. By the way, the treatment has been shown to be well tolerated, and no new safety signals were identified.

Vibostolimab is a humanized anti-TIGIT IgG1 antibody developed by Merck. Applied as a monotherapy or in combination with the anti-PD-1 antibody pembrolizumab, it showed a manageable safety profile and promising therapeutic potential in first-line treatment patients [170]. However, only modest antitumor activity was achieved in NSCLC patients refractory to anti-PD-1/PD-L1 immunotherapy. The coformulation of pembrolizumab/vibostolimab is currently being evaluated in the phase III clinical trial KEYVIBE (NCT04738487) in PD-L1 positive metastatic NSCLC as first-line treatment compared to pembrolizumab alone [171].

### 3.3. CD137

CD137 belongs to the TNF receptor superfamily (TNFRSF). It is expressed on CD4 and CD8 T cells shortly after antigen exposure [172], and its ligation, together with the TCR, delivers co-stimulatory signals that promote T cell expansion with memory phenotypes and stronger effector capacities in terms of cytotoxic capacities and cytokine production [173]. Interestingly, these effects are particularly pronounced on CD8 T cells rather than on CD4 lymphocytes [173]. Apart from CD137, other costimulatory receptors of the TNFRSF family, such as OX40, GITR, or CD27, have been targeted with agonistic antibodies to promote anti-tumor T cell responses. However, most of them have shown weak or not durable responses according to preliminary results in phase I clinical trials when administered as monotherapies [174,175,176,177,178,179,180,181,182,183,184,185,186,187]. Apart from toxicity-related concerns, the determination of optimal dosing schedules seems to be of great importance, since a minimum half-life is required for T cell engagement, while persistent stimulation may lead to T cell exhaustion. CD137 appears to be the most robust target among the TNFRSF costimulatory receptors under current clinical investigation. Two main agonistic antibodies have been developed up to date, urelumab, and utomilumab.

Urelumab (BMS-663513) is a humanized IgG4 agonistic antibody to CD137 developed by Bristol-Myers Squibb. The first results from clinical trials have demonstrated clinical responses in patients with advanced solid tumors and lymphomas [188,189]. However, hepatotoxicity concerns arose associated with required doses above the maximum tolerated limits [190]. Utomilumab (PF-05082566) is a humanized IgG2 agonistic antibody to CD137 developed by Pfizer, that has been proved to be well tolerated in a phase I clinical trial (NCT01307267) [191], although limited efficacy was achieved. Due to security profiles and efficacy limitation issues, the combination of CD137 agonistic antibodies with other immuno oncology agents is being analyzed in order to achieve sufficient therapeutic effects [192,193,194]. Bispecific antibodies for the simultaneous targeting of CD137 and tumour-associated antigens (TAA) are attracting increasing attention as a strategy to reduce off-target toxicities (Appendix A).

### 3.4. ICOS

Inducible T cell costimulatory (ICOS) is an activating costimulatory checkpoint expressed on activated T cells. Its unique ligand, ICOSL, is expressed by APCs and somatic cells, including tumor cells [195,196,197,198]. The ICOS/ICOSL pathway represents an attractive target for immunotherapy, due to its dual role in the context of cancer. While its ligation promotes an increase in the Teff/Treg ratio in the tumor microenvironment and differentiation of Th1 TILs [199,200], its sustained activation induces immunosuppression mediated by Tregs [201,202,203,204], thus displaying both pro- and anti-tumorigenic potential. Indeed, both agonistic and blocking antibodies that target this pathway are being evaluated in clinical trials (Appendix A).

Preliminary results from phase I clinical trials show that agonistic monoclonal antibodies targeting ICOS rendered promising clinical responses with good safety and tolerability profiles, whereas ICOS antagonistic antibodies, aimed at ICOS expressing-Tregs abrogation, did not achieve remarkable response rates [205,206,207,208]. Interestingly, preclinical results demonstrated a strong synergy between ICOS antibodies and other ICB agents, such as CTLA-4 or PD-1/PD-L1 [209]. Therefore, most ongoing clinical trials are evaluating combinations of anti-ICOS agonistic and antagonistic antibodies with conventional ICB therapies.

Vopratelimab (JTX-2011) is the most advanced ICOS-agonistic antibody up to date. It is an IgG1 monoclonal antibody developed by Jounce Therapeutics. The first-in-human phase I/II ICONIC trial (NCT0290422) revealed that this antibody boosts the activation and proliferation of primed CD4 T effector cells and leads to improved clinical outcomes in patients treated with this antibody both as a monotherapy and in combination with the anti-PD-1 inhibitor nivolumab [210,211]. Following promising results from this phase I clinical trial with no adverse safety and tolerability concerns, it is currently being analyzed in the SELECT phase II trial (NCT04549025) in combination with another PD-1 inhibitor (pimivalimab, JTX-4014) in NSCLC patients [212].

## 4. Conclusions

ICIs have revolutionized oncology medical practice since the FDA approval of the first ICI 11 years ago. However, one of the main current challenges in oncology is that many patients do not respond to treatment due to intrinsic and extrinsic factors, which is a major clinical problem. In light of this, LAG-3 is one of the most important next-generation immune checkpoint molecules, playing a similar role to PD-1 and CTLA-4 inhibitory molecules. LAG-3 plays a critical role in regulating T cell homeostasis, acting as an inhibitor of T cell activation, proliferation, cytokine secretion, and effector functions. Thus, 108 interventional clinical trials are evaluating 19 different LAG-3 targeting molecules in phases I, I/II, II, II/III, and III, demonstrating strong positive results, including promising bispecific molecules targeting LAG-3 with other immune checkpoint inhibitors, especially with anti-PD-1 blockers.

The recent approval of Opdualag, a novel immune checkpoint inhibitor fixed-dose treatment combination of nivolumab (anti-PD-1) and relatlimab (anti-LAG-3) developed by Bristol Myers Squibb for the treatment of first-line unresectable melanoma or melanoma that has spread (advanced melanoma) is a significant large milestone in the clinical landscape of cancer treatment. This LAG-3 and PD-1 co-blockade has proven to be safe and effective, providing greater benefit in terms of PFS when compared to PD-1 blockade alone, with manageable toxicity profiles. Remarkably, it more than doubles the median PFS when compared to nivolumab monotherapy (10.1 months versus 4.6 months), demonstrating to be safe and effective. This regulatory approval opens a door to the entry of more promising LAG-3 targeting molecules in the clinical practice market, and to an extension of Opdualag indications to other malignancies, supporting the accumulating evidence that indicates that LAG-3 will play a pivotal role in cancer treatment equivalent to current anti-PD-1/anti-PD-L1 treatments.

## Figures and Tables

**Figure 1 cells-11-02351-f001:**
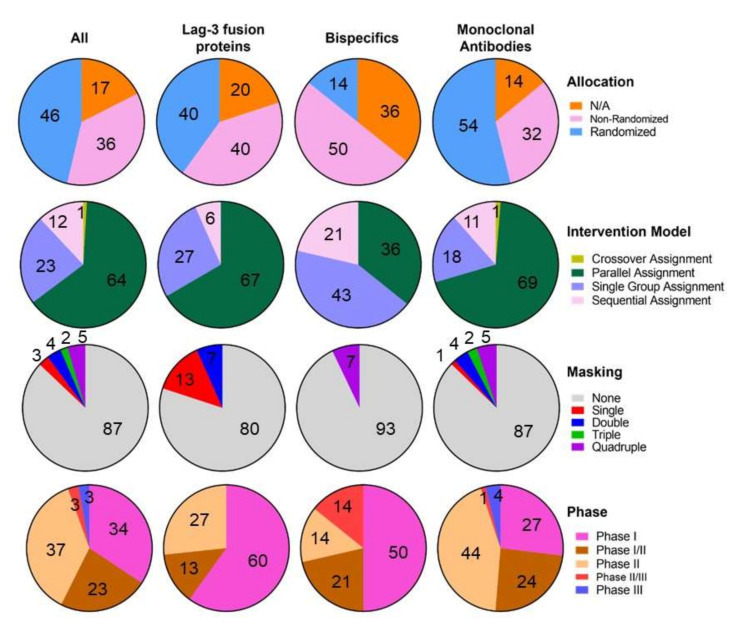
LAG-3-targeted therapy clinical landscape allocation, intervention models and masking of LAG-3 targeted molecules including LAG-3 fusion proteins, bispecific molecules and monoclonal antibodies (https://clinicaltrials.gov/, accessed on 29 June 2022). Percentages are indicated within the graphs.

**Figure 2 cells-11-02351-f002:**
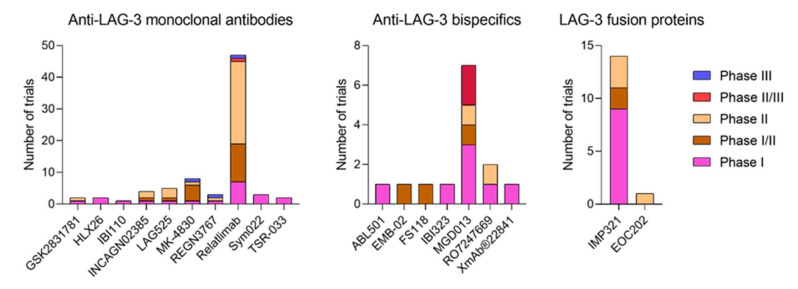
LAG-3-targeted therapy clinical landscape of phases for anti-LAG-3 monoclonal antibodies, bispecific molecules and fusion proteins clinically developed (https://clinicaltrials.gov/, accessed on 29 June 2022).

**Table 1 cells-11-02351-t001:** Summary of the most common AEs and SAEs associated with the treatment with Opdualag compared to Nivolumab alone in the RELATIVITY-047 clinical trial (NCT03470922).

		Opdualag (*n* = 355)	Nivolumab (*n* = 359)
**AEs**	Musculoskeletal pain	54.6%	33.1%
	Fatigue	29.3%	20.6%
	Asthenia	13.5%	9.2%
	Pyrexia	12.4%	9.2%
	Headache	18.0%	13.1%
	Cough	14.1%	10.6%
	Rash	17.4%	13.6%
	Pruritus	24.8%	17.3%
	Diarrhea	23.1%	17.3%
	Nausea	17.7%	16.4%
	Constipation	11.0%	7.0%
	Decreased appetite	15.5%	7.5%
	Anaemia	14.1%	10.3%
	Increased AST	9.9%	4.7%
	Increased ALT	10.1%	5.85%
	Hypothyroidism	16.3%	13.1%
**SAEs**	Anaemia	1.4%	1.1%
	Acute myocardial infarction	1.1%	0.6%
	Myocarditis	1.1%	0.3%
	Adrenal insufficiency	1.4%	0.0%
	Colitis	1.4%	0.3%
	Diarrhea	1.1%	0.8%
	General health deterioration	0.6%	1.7%
	Pyrexia	0.9%	1.4 %
	Pneumonia	1.4%	0.8%
	Urinary tract infection	0.9%	1.7%
	Back pain	1.1%	0.6%
	Malignant neoplasm progression	11.0%	13.1%
	Metastases to central nervous system	1.1%	0.8%
	Pneumonitis	1.1%	0.3%

## Data Availability

Not applicable.

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
