# Peer review of "Cutting-Edge: Preclinical and Clinical Development of the First Approved Lag-3 Inhibitor"

_cells, 2022, doi:10.3390/cells11152351_

Round 1
Reviewer 1 Report
Excellent and comprehensive review of a large number of clinical trials. A very interested reader may possibly be interested in whether combination anti-PD1 and anti-LAG is more toxic. A table comparing side effect profile of single agent vs dual therapy could be helpful. Instead of burying side effects in the text, a table may possibly make the paper a better contribution
Author Response
Dear Reviewer,
many thanks for your valuable suggestion.
In the revised version of the manuscript, we have removed the description of opdualag's side effects from the main text and we have replaced it with a table that establishes a comparison of the single versus dual therapy, thus clarifying the toxicity profiles of both.
Reviewer 2 Report
The authors summarized clinical research of LAG-3 inhibitors for cancer and perspective on cancer or other diseases. Although the manuscript is quite well written and clearly presented, several modifications are needed as below.
Abstract
Line 17: “Immune checkpoint inhibitors” should be singular since ICIs were described in several sentences.
Line 18, 19, 23: LAG-3, PD-1, CTLA-4, FDA should be spelled out.
Main text
Line: 41: “Immune checkpoint inhibitors” should be singular and PD-1, PD-L1, CTLA-4, and LAG-3 should be spelled out. Because Abstract is independent of the main text.
Line 62: LAG3 is correctly LAG-3.
Line 81: MHC-II should be spelled out.
Line93: 2.1 is correctly 2.2.
Line 115: 28000 is correctly 28,000.
Line 219 “2.1.1 Anti-LAG-3 monoclonal antibodies” is correct “2.2.1 Anti-LAG-3 monoclonal antibodies”? Is the heading number after this also correct?
3.1.2. Anti-LAG-3 bispecific antibodies > 2.2.2
3.1.3, LAG-3 fusion proteins > 2.2.3
3.1.3. Anti-LAG-3 CAR-T cells > 2.2.4
2.2. Opdualag and its pathway towards the clinic > 2.3
Line: 662: Insert space into the top of the paragraph. “Immune checkpoint inhibitors (ICI)” should be changed to “ICIs” since this word had already been spelled out.
Figure 1.
The percentage should be shown.
Line 182: “LAG-3 targeted molecules (LAG-3 fusion proteins, bispecific molecules and monoclonal antibodies)” would be changed to “LAG-3 targeted molecules including LAG-3 fusion proteins, bispecific molecules, and monoclonal antibodies”
Author Response
Dear Reviewer,
Many thanks for you valuable suggestions.
Please find below our point-by-point response to your comments:
- We have unified all the abbreviations throughout the main text.
- We have spelled out all the abbreviations separately in the Abstract section and the main text.
- We have corrected the sections heading numbers.
- We have added the percentages to Figure 1 and corrected the figure caption.